# QFR Predicts the Incidence of Long-Term Adverse Events in Patients with Suspected CAD: Feasibility and Reproducibility of the Method

**DOI:** 10.3390/jcm9010220

**Published:** 2020-01-14

**Authors:** Andrea Buono, Annika Mühlenhaus, Tabitha Schäfer, Ann-Kristin Trieb, Julian Schmeißer, Franziska Koppe, Thomas Münzel, Remzi Anadol, Tommaso Gori

**Affiliations:** Kardiologie I, Universitätsmedizin Mainz and DZHK Standort Rhein-Main, 55128 Mainz, Germany; andrebuo@hotmail.com (A.B.); annika.muehlenhaus@web.de (A.M.); tabitha.schaefer@gmx.de (T.S.); ann-kristin.trieb@gmx.de (A.-K.T.); julian.schmeisser@t-online.de (J.S.); franziska.koppe@unimedizin-mainz.de (F.K.); tmuenzel@uni-mainz.de (T.M.); remzi.anadol@unimedizin-mainz.de (R.A.)

**Keywords:** fractional flow reserve, coronary artery disease, coronary interventions, Quantitative flow reserve

## Abstract

AIMS: We evaluate feasibility and reproducibility of post hoc quantitative flow ratio (QFR) measurements and their prognostic predictive power during long-term follow-up. METHODS AND RESULTS: Between 2010 and 2012, 167 patients without angiographic evidence of significant stenoses were enrolled in a prospective registry. Of these patients, 96% presented 7 years follow-up data. QFR was measured post hoc by three certified investigators. QFR analysis was feasible in 71% of left anterior descending (LAD), 72% of left circumflex (LCX), and 61% of right (RCA) coronaries for a total of 350 measurements repeated in triplicate. Coefficients of variation were 2.1% for RCA and LCX, and 2.8% for the LAD (quartile coefficients of dispersion respectively 1.5, 1.4, and 1.3). QFR ≤0.80 was recorded in 25 patients (27 vessels, in 74% of the cases LAD). A total of 86 major adverse cardiovascular and cerebrovascular events were observed in 76 patients. QFR ≤0.80 in at least one of the three vessels was the strongest predictor of events (HR 3.14, 95%CI 1.78–5.54, *p* = 0.0001). This association was maintained in several sensitivity analyses. CONCLUSIONS: QFR reproducibility is acceptable, even when analysis is performed post hoc. A pathological QFR is not rare in patients without angiographic evidence of significant stenosis and is a predictor of incident events during long-term follow-up. Condensed Abstract: In a post hoc analysis of 167 patients without evidence of angiographic significant stenosis, the presence of QFR value ≤0.80 in at least one of the three coronary vessels showed to be the strongest predictor of major adverse cardiovascular and cerebrovascular events during long-term follow-up. QFR reproducibility have been shown to be acceptable among experienced operators.

## 1. Introduction

Invasive coronary angiography is the most commonly used method for the identification of coronary artery disease (CAD) in patients with high suspicion. However, angiography alone does not accurately reflect the hemodynamic relevance of coronary lesions. Several tools have been developed to overcome this limitation, among which fractional flow reserve (FFR) and instantaneous wave-free ratio (iFR) represent the most frequently adopted. Given their reliability in the prediction of cardiovascular events and consequently in guiding the decision whether to perform percutaneous coronary intervention (PCI) [1,2,3,4,5,6,7,8], the use of these methods is recommended as Class IA in the current guidelines on myocardial revascularization [9]. Despite the indisputable benefit of these functional indexes, their use increases the invasiveness and the cost of procedure, and their penetration remains unfortunately low.

In the last years, a novel approach enabling rapid computation of the hemodynamic relevance of a stenosis based on three-dimensional quantitative coronary angiography and fluid dynamic algorithms has been developed; the index derived from this analysis, named quantitative flow ratio (QFR), has been validated against FFR in multiple studies [10,11,12,13]. QFR has shown to have a significantly higher sensitivity and specificity compared with quantitative coronary analysis (QCA) [14]. Increasing evidence supports its applicability in different conditions. In fact, its usefulness has been demonstrated not only in the context of native coronary artery stenosis, including small vessel disease [15], but also in particular lesion settings, such as in-stent restenosis [16]. Moreover, its use can be applied immediately after PCI, in order to confirm the hemodynamic results of the revascularization and eventually to guide further treatment, as recently demonstrated in the HAWKEYE study [17]. QFR is also a valuable tool in the context of acute coronary syndrome: an incomplete revascularization detected by QFR has been associated with a higher rate of cardiovascular events in an ST elevation myocardial infarction (STEMI) population [18].

Despite these promising data, several issues remain to be clarified before QFR is introduced in clinical practice. First, QFR requires high-quality end-diastolic images of the whole segment under exam in at least two angiographic views with at least 25° angle separation, which might not be available from standard routine angiographies not performed for the purpose of QFR analysis. Second, it remains unclear to what extent QFR allows identifying lesions requiring treatment that are otherwise not found at angiography. Finally, evidence concerning the prognostic predictive power of QFR during long-term follow-up is emergent.

## 2. Methods

### 2.1. Goal of the Study and Study Design

We set out to assess the feasibility and reproducibility of QFR measurements performed post hoc in standard angiographic exams. Further, the goal of the study was to investigate whether the assessment of QFR in patients with an angiographic diagnosis of nonrelevant coronary artery disease allows identification of hemodynamically relevant stenosis that were missed at angiography, and whether vessels with abnormal QFR left untreated are associated with worse patients′ prognosis.

This single-center registry enrolled consecutive patients with suspected coronary artery disease in whom the presence of relevant stenosis was excluded at angiography. Follow-up data were collected prospectively. The study complied with the Declaration of Helsinki for investigation in human beings and was approved by the ethics committee of the Landesärztekammer Rheinland-Pfalz. All patients gave written informed consent. The study is registered in clinicaltrials.gov as NCT01787370.

### 2.2. Study Population and Endpoint Definition

Consecutive patients (≥18 years old) with suspected coronary artery disease (stable angina or stable angina equivalent) undergoing coronary angiography were consecutively enrolled between January 2010 and February 2012 in the Flow-Mec registry, a study designed to prospectively investigate novel early and long-term determinants of cardiac and cerebrovascular prognosis [19]. From this database, consecutive patients with mild to moderate (10–70%) lesions at angiography were included in the current analysis: the presence of a stenosis judged to be significant at angiography and treated ad hoc or scheduled for subsequent revascularization, a previous history of coronary artery bypass grafting (CABG), the presence of unstable symptoms (worsening angina or rest angina within one month), a myocardial infarction (MI) episode or cardiac catheterization within three months before angiography and presence of chronic total occlusions were exclusion criteria.

The primary endpoint of the Flow-Mec study was the incidence of major adverse cardiovascular and cerebrovascular events (MACCE), defined as a composite of all-cause mortality, MI (both non ST-segment elevation and ST-segment elevation MI), cerebrovascular stroke, and any incident coronary artery revascularization (both surgical and/or percutaneous). Secondary endpoints included target vessel failure (TVF), defined as composite of cardiovascular death, target vessel myocardial infarction, and target vessel revascularization, as well as each separate component of the composite endpoint.

The incidence of events was assessed with in-person or telephone visits. All events were validated against original clinical documents.

### 2.3. Data Management and Analysis

QFR was measured from original coronary angiography DICOM files stored in the hospital′s picture archiving and communication system. Angiograms were performed using a Philips AlluraClarity (Philips Medical Systems, Amsterdam, the Netherlands) and stored in Xcelera (Philips medical systems). Three certified investigators (A.M., T.S., and A.T.) independently analyzed each coronary vessel using QFR Medis Suite system (Medis, Leiden, the Netherlands) following the manufacturer’s instructions. Investigators were unaware of the patients’ follow-up data.

### 2.4. Statistics

Descriptive statistics are reported as mean ± SD, median (interquartile range), or frequencies (%), as appropriate. Normal distribution was tested by inspection of the Q–Q plots. Coefficients of variation and quartile coefficients of dispersion among the three investigators′ measurements were determined to assess the repeatability of the measurements.

The primary endpoint of the study was to test whether QFR is a predictor of MACCE during 7 years follow-up in multivariable analysis. A list of all prespecified parameters included in univariate analysis is provided in Table 1. Parameters with a univariate *p* < 0.05 were entered in multivariable analysis.

Pairwise comparisons were made with Student *t*-test or Chi-squared tests, as appropriate. Hazard ratios (HR) were calculated with Cox proportional hazards regression. For survival analysis, Kaplan–Meyer survival curves were drawn. A two-sided *p* value of <0.05 was considered significant for the primary endpoint, all other analyses are exploratory. Statistical analysis was performed with MedCalc version 13.0 (Mariakerke, Belgium).

## 3. Results

### 3.1. Feasibility and Clinical Characteristics

The study flow is shown in Figure 1. A total of 167 patients meeting the inclusion and exclusion criteria were identified from the Flow-Mec database, and their characteristics are presented in Table 1. The mean age of study population was 65 ± 11.3 years, 70% patients were male, and 17.3% had diabetes. Of the 501 vessels, 151 (30%) could not be analyzed due to absence of adequate high-quality end-diastolic frames containing the whole vessel length in at least two projections and separated by an angle >25°, without overlap at the lesion segment of interest, excessive foreshortening, or insufficient contrast filling. In 11 patients, none of the three vessels could be analyzed. In sum, QFR of the left anterior descending (LAD) could be analyzed in 122 (73%) patients, QFR of the left circumflex (LCX) could be analyzed in 123 (74%), and the right coronary artery (RCA) could be analyzed in 105 (63%).

### 3.2. Reproducibility

Each QFR analysis was performed by three independent certified operators for a total of 1050 measurements. The mountain plots are presented in Appendix A. The coefficient of variation was 2.8% for QFR measurements in the LAD, 2.1% for the LCX and RCA. The corresponding quartile coefficients of dispersion were 1.5, 1.4, and 1.3.

### 3.3. QFR Analysis

Examples of QFR analyses are presented in Figure 2. The mean QFR values were 0.91 ± 0.07 for the LAD, 0.96 ± 0.04 for the LCX, and 0.96 ± 0.05 for the RCA. There were 27 vessels (7.7%) with QFR ≤0.80 (LAD in 20 patients, LCX in 3, and RCA in 4). Two patients had a pathological QFR in two vessels (LAD and LCX in one patient, LCX and RCA in the other). The remaining 131 patients had QFR >0.80 in all vessels analyzed. Patients with at least one vessel with QFR ≤0.80 had more frequently a history of PCI (64% vs. 39.7%, *p* = 0.04). Otherwise, there was no difference between patients with at least one vessel with QFR ≤0.80 and those without pathological QFR values. Table 2 summarizes the main angiographic findings. The degree of coronary stenosis at quantitative angiography (QCA) was significantly higher in the group of patients with pathological QFR, both in terms of diameter (DS%) and area stenosis (AS%). Despite these differences, no lesion in either group was angiographically severe (>75%) at QCA analysis. Lesion length was longer in the LCX (24.3 ± 16.0 vs. 14.6 ± 8.2 mm, *p* = 0.0001), and reference vessel diameter was smaller in the LAD (2.3 ± 0.6 vs. 2.6 ± 0.7 mm, *p* = 0.04) in patients with QFR ≤0.80. QFR analysis showed differences for all three vessels.

### 3.4. Follow-Up

Seven years follow-up data were available in 96% of the patients.

### 3.5. MACCE

A total of 86 events in 76 patients were observed at 2334 (1846–2580) days. There were 32 deaths (10 of cardiac origin), 5 cerebrovascular strokes, 15 new MI, and 34 repeated revascularizations (30 PCI and 4 CABG).

In univariate analysis (Table 3), older age (HR 1.023, 95%CI 1–1.04, *p* = 0.03), male sex (HR 2.24, 95%CI 1.27–3.94, *p* = 0.005), chronic kidney disease (CKD) (HR 2.27, 95%CI 1.09–4.72, *p* = 0.03), a previous history of PCI (HR 2.03, 95%CI 1.28–3.20, *p* = 0.0025), and the presence of QFR ≤0.80 in at least one of the three vessels (HR 3.42, 95%CI 2.06–5.67, *p* < 0.0001) were predictors of MACCE. Among QCA parameters, AS% and DS%, but not RVDs and lesion length, showed an association with MACCE.

Seven variables were entered in multivariate analysis (Table 4). Only a previous history of PCI (HR 1.90, 95%CI 1.15–3.15, *p* = 0.0134), chronic kidney disease (HR 2.81, 95%CI 1.26–6.26, *p* = 0.0118), and the presence of QFR ≤0.80 in at least one of the three vessel (HR 3.14, 95%CI 1.78–5.54, *p* = 0.0001) were predictors of long-term MACCE incidence. The Kaplan–Meier survival curves describing the impact of pathological QFR are shown in Figure 3. The association between QFR and events was driven mostly by repeat revascularizations (HR 0.22, 95%CI 0.03–0.23, *p* < 0.0001), while the association with myocardial infarction (HR 0.42, 95%CI 0.07–1.39, *p* = 0.1258), stroke (*p* = 0.432), and death (HR 0.85, 95%CI 0.30–2.31, *p* = 0.7335) were not significant.

### 3.6. Secondary Endpoints

The predictive role of QFR ≤0.80 was maintained in a subanalysis limited to patients without a prior history of revascularization. In these patients, QFR ≤0.80 was associated with a 3.24-fold increase in risk of long-term MACCE in this subgroup of patients (HR 3.24, 95%CI 1.40–7.48, *p* = 0.006). The corresponding Kaplan–Meier curve is presented in Appendix A. Similarly, a sensitivity analysis excluding patients with kidney failure confirmed the predictive power of QFR (HR 4.26, 95%CI 2.52–7.20, *p* < 0.001).

Figure 4 shows the receiver operating characteristic (ROC) curves for QFR in individual coronary arteries. This analysis showed a stronger association for the QFR values assessed in the LAD and the RCA (area under the curve of the LAD: 0.65, 95%CI 0.56–0.73, *p* = 0.002; RCA: 0.67; 95%CI 0.57–0.76; *p* = 0.001), whereas the QFR measured in the LCX showed a weaker association (area under the curve 0.54, 95%CI 0.44–0.63, *p* = 0.50). The thresholds associated with the best combination of sensitivity and specificity were QFR ≤0.85 in the LAD (sensitivity: 42.1% (95%CI 29.1–55.9%), specificity: 84.1% (95%CI 72.7–92.1%), positive predictive value 2.65, negative predictive value 0.69), while a value ≤0.80 demonstrated a very high specificity but a low sensitivity (sensitivity: 29.8% (95%CI 18.4–43.4%), specificity: 95.2% (95%CI 86.7–99%), positive predictive value 6.26, negative predictive value 0.74). In analogy, QFR ≤0.96 in the RCA was associated with the best combination of sensitivity and specificity (sensitivity 49%, 95%CI 34.8–63.4%; specificity 79.6%, 95%CI 66.5–89.4%) positive predictive value 2.41, negative predictive value 0.64).

During follow-up, 25 TLFs of the LAD occurred. Incident TLF during follow-up identified patients with a significantly lower value of LAD QFR at index (0.83, 95%CI 0.71–0.91 vs. 0.90, 95%CI 0.86–0.97; *p* = 0.002). Figure 5 shows the Kaplan–Meier curves for cut-off values of LAD QFR ≤0.80 and ≤0.90. With both cut-offs, a pathological QFR of the LAD was associated with incident TLF.

Finally, Appendix A show the curves describing the association of QFR ≤0.80 and QFR ≤0.90 with individual endpoints. For both QFR thresholds, there was an association with incident revascularizations (*p* < 0.0001 for QFR ≤0.80, *p* = 0.0013 for QFR ≤0.90), but not for death or incident myocardial infarction.

## 4. Discussion

We report on the feasibility, reproducibility, and predictive power of QFR in a group of patients enrolled in the Flow-Mec study, a prospective registry on the predictors of ischemic events in patients undergoing coronary angiography. The main findings of this study include the following: 1. QFR assessments are feasible in only 70% of coronary angiograms when these are not performed for the purpose (and therefore not respecting the specific criteria) of QFR measurements. 2. The reproducibility of these assessments is good, and similar to that reported in previous QFR studies. 3. Findings of pathological QFR are not rare (7.7% of all vessels measured) and are more frequent in the LAD than in LCX and RCA. In our population, 16% of patients affected by intermediate coronary artery stenosis judged angiographically not significant presented a pathological QFR value at off-line analysis. This aspect corroborates the importance to perform a more routinely use of functional tools in the context of intermediate coronary stenosis. 4. The presence of a QFR ≤0.80 is associated with a 3-fold increase in the risk of major adverse cardiovascular and cerebrovascular events, principally driven by an increased risk of target vessel failure and revascularization. Importantly, the survival curves diverge already during the first months after the index catheterization. In our population, also CKD (HR 2.81, 95%CI 1.26–6.26, *p* = 0.01) and a previous history of PCI (HR 1.90, 95%CI 1.15–3.15, *p* = 0.01) have been associated with a higher risk of long-term MACCE, despite their predictive values being lower compared with QFR ones (HR 3.14, 95%CI 1.78–5.54, *p* = 0.0001). Interestingly, in the meantime, other variables such as obesity and received anti-ischemic treatment did not correlate with the risk of MACCE, suggesting that prognosis in patients affected by intermediate coronary stenosis is determined by multiple factors, related not only to a patient’s classical clinical risk profile and medications, but also to plaque’s characteristics and rheological aspects.

Invasive methods to assess the hemodynamic significance of coronary artery stenoses are associated with improved short- and long-term outcomes, principally driven by a reduced incidence of revascularization and target vessel myocardial infarction [1,2,3]. Despite their advantages, the penetration of these invasive techniques remains low due to cost or logistic considerations, limited access, and a perceived increased risk associated with intracoronary procedures and the administration of vasodilators. Further, an intrinsic disadvantage of these invasive procedures is that they need to be performed at the time of catheterization, and a post hoc reassessment of the hemodynamic relevance of a stenosis is clearly not possible. QFR assessment has the advantage of a reduced invasiveness, no additional risks beyond those of invasive catheterization, and that it allows off-line re-evaluation of stenosis any time after conclusion of the angiogram. Evidence from a recent retrospective study demonstrates that QFR predicts the incidence of major adverse cardiovascular events (MACE) during a 2.2 years follow-up [20]. The current manuscript expands on these data, providing evidence of the feasibility, reproducibility, and clinical relevance of QFR data during a 7 years follow-up. Taken together, the current data validate the use of QFR also for off-line, post hoc decisions for the interpretation of coronary angiograms, even though the feasibility of the measurements is relatively low (70%) in angiograms not performed according to the specific QFR requirements. Our data reproduce the results of the large FFR trials, providing further evidence that lesions and patients at risk may be identified by measuring the hemodynamic relevance of coronary stenoses even when, based on angiography alone, a decision against intervention has been taken.

### Limitations

Our study has several limitations. First, angiograms were performed between 2010 and 2012 (when QFR was not yet available), and although follow-up was prospective, Digital Imaging and Comunication in Medicine (DICOM) movies were retrospectively analyzed. Although this methodological limitation needs to be acknowledged, QFR was not available at the time of patient inclusion, and it would have been ethically unacceptable to prospectively study the outcome of patients in whom a pathological QFR had been found. The feasibility of post hoc QFR measurements was one of the endpoints of the study. QFR analysis was not possible in about 30% of the vessels, and no conclusion can be taken regarding the presence of relevant stenoses in these vessels and how these affected patient prognosis. Also, side branches were not measured, and QFR has not been validated for ostial lesions. Second, some patients had a history of prior PCI, which also may influence the results and the association between events and stenosis at index. A sensitivity analysis performed after exclusion of these patients confirmed however the main findings of the study. QFR has not been validated in by-pass recipients, and these patients were therefore excluded from the present analysis. The prespecified primary endpoint of Flow-Mec was MACCE, while previous FFR/iFR studies focused on MACE or TLF. Exploratory analyses using these endpoints were however consistent with the main results. Finally, adherence to medical therapy during follow-up was not assessed.

## 5. Conclusions

QFR analysis is feasible in 70% of routinely performed angiograms. QFR is a reproducible and effective diagnostic tool that improves, without additional risks, the ability of angiography to identify significant coronary artery stenoses. A value of QFR ≤0.80 detected in at least one coronary vessel predicts a 3-fold higher risk to develop a major cardiovascular and cerebrovascular adverse event during long-term follow-up.

## 6. Impact on Daily Practice

QFR is a valuable and reproducible tool to assess hemodynamic severity of coronary artery stenosis. QFR ≤0.80 in at least one of the three coronary vessels should be considered as a prognostic risk factor to develop long-term follow-up major adverse cardiovascular and cerebrovascular events.

## Figures and Tables

**Figure 1 jcm-09-00220-f001:**
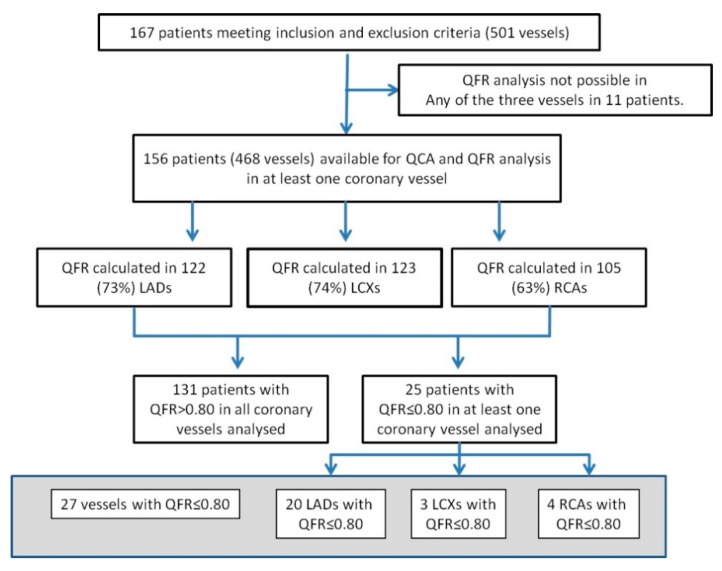
Study Flow Chart. QFR: Quantitative Flow Reserve; QCA: Quantitative Coronary Analysis; LAD: Left Anterior Descending Artery; LCX: Left Circumflex Artery; RCA: Right Coronary Artery.

**Figure 2 jcm-09-00220-f002:**
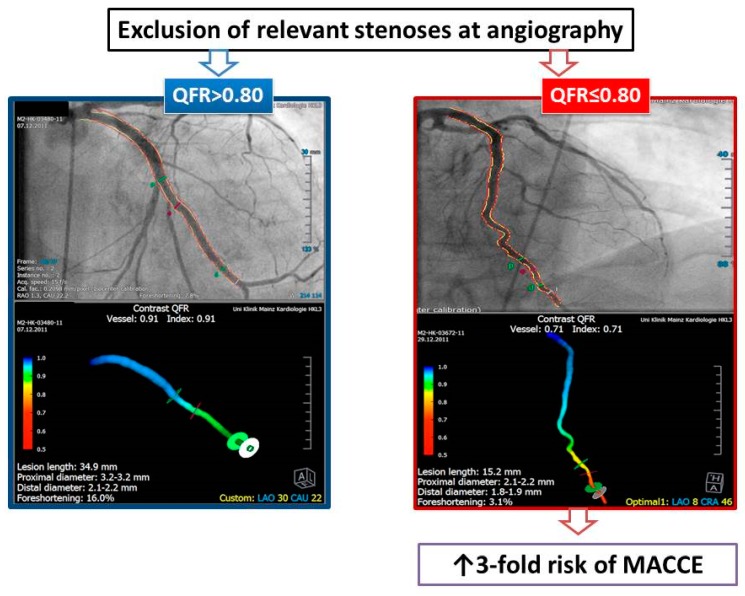
QFR analysis and take home message. Detection of QRF ≤0.80 in at least one coronary vessel is associated with a 3-fold higher risk to develop an adverse cardiovascular and cerebrovascular event during a long-term follow-up. MACCE: major adverse cardiovascular and cerebrovascular events.

**Figure 3 jcm-09-00220-f003:**
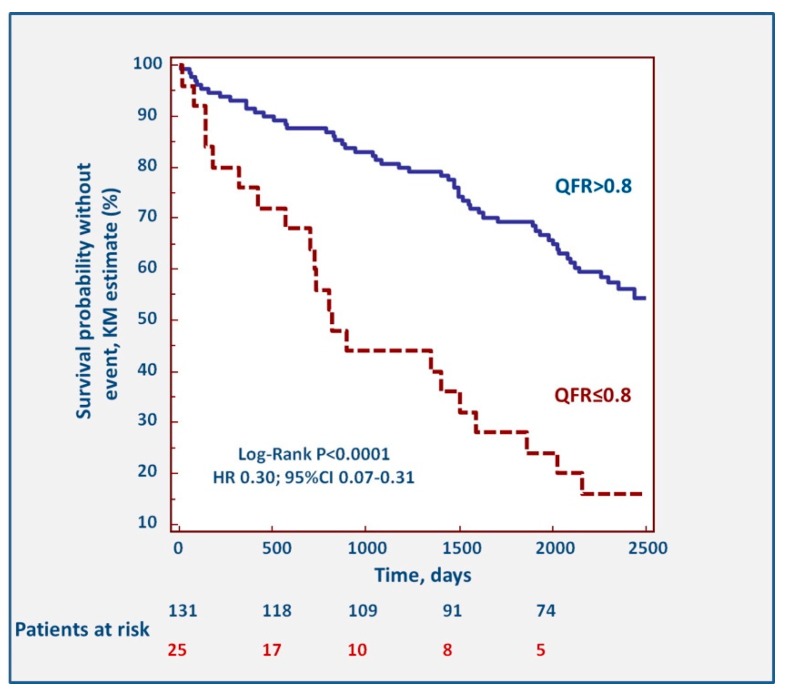
Kaplan–Meier survival curves of patients presenting with QFR >0.80 and ≤0.80.

**Figure 4 jcm-09-00220-f004:**
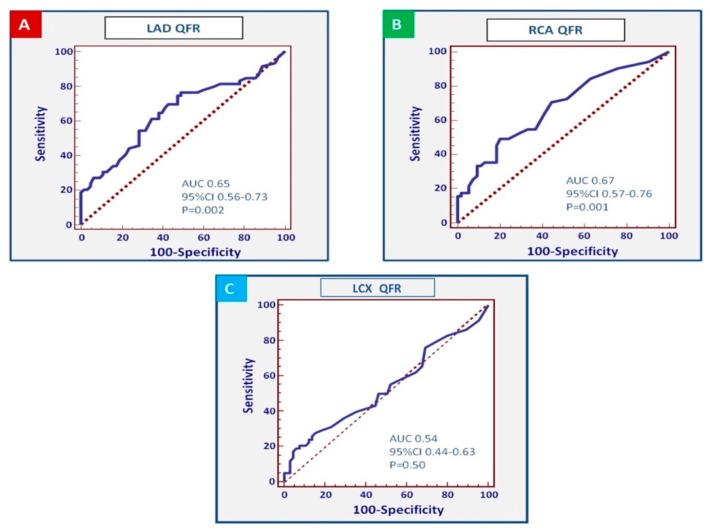
ROC (receiver operating characteristic) curves of LAD QFR (**A**), RCA QFR (**B**), and LCX QFR (**C**). AUC: area under the curve.

**Figure 5 jcm-09-00220-f005:**
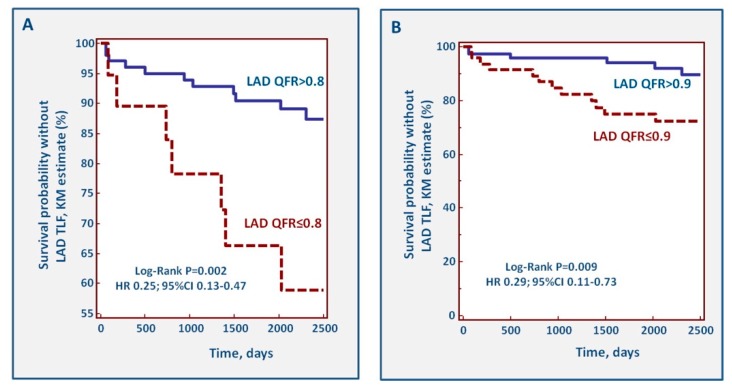
Kaplan–Meier curves for LAD-TLF assuming a cut-off LAD QFR value ≤0.80 (**A**) and ≤ 0.90 (**B**).

**Table 1 jcm-09-00220-t001:** Study population characteristics. Comparison between population subgroups divided according to QFR (quantitative flow ratio) value (QFR > 0.80 in all the analyzed vessel and ≤0.80 in at least one vessel).

	QRF > 0.80 (131)	QFR ≤ 0.80 (25)	All Patients	*p*-Value
Age, years ± SD	65.1 ± 11.3	66 ± 11.5	65.1 ± 11.3	0.70
Male sex, *n* (%)	87 (66.4)	21 (84.0)	108 (70.0)	0.13
Atrial Fibrillation, *n* (%)	20 (15.3)	7 (28.0)	27 (17.3)	0.22
Chronic obstructive pulmonary disease, *n* (%)	12 (9.1)	5 (20.0)	17 (10.9)	0.22
Diabetes mellitus, *n* (%)	21 (16)	6 (24.0)	27 (17.3)	0.51
Family history of CAD, *n* (%)	36 (27)	4 (16.0)	40 (25.6)	0.37
Dyslipidemia, *n* (%)	72 (55)	16 (64.0)	88 (56.4)	0.56
Hypertension, *n* (%)	104 (79.4)	21 (84.0)	125 (80.1)	0.78
Chronic kidney disease, *n* (%)	9 (6.9)	1 (4.0)	10 (6.4)	0.92
Obesity, *n* (%)	42 (32)	7 (28.0)	49 (31.4)	0.85
Smoking, *n* (%)	64 (48.8)	12 (48.0)	76 (48.7)	0.92
Severe valvular heart disease, *n* (%)	7 (5.3)	0 (0.0)	7 (4.5)	-
Previous PCI, *n* (%)	52 (39.7)	16 (64.0)	68 (43.6)	0.04
Previous MI, *n* (%)	32 (24.4)	5 (20.0)	37 (23.7)	0.82
Left ventricular ejection fraction <30%, *n* (%)	5 (3.8)	2 (8.0)	7 (4.5)	0.60
ACEi and/or ARBs, *n* (%)	90 (68.7)	20 (80)	110 (70.5)	0.40
Aspirin, *n* (%)	89 (67.9)	19 (76)	108 (69.2)	0.56
Beta-Blockers, *n* (%)	79 (60.3)	18 (72)	97 (62.1)	0.36
Calcium Channel Blockers, *n* (%)	31 (23.7)	8 (32.0)	39 (25)	0.49
Nitrates, *n* (%)	21 (16)	4 (16.0)	25 (16.1)	0.78
Statin, *n* (%)	70 (53.4)	19 (76.0)	89 (57.4)	0.07

CAD: coronary artery disease. ACEi: angiotensin-converting enzyme inhibitors; ARB: angiotensin receptor blockers. Chronic kidney disease: GFR < 60 mL/min. CABG: coronary by-pass. MI: myocardial infarction; PCI: percutaneous coronary intervention. SD: Standard deviation.

**Table 2 jcm-09-00220-t002:** Angiographic characteristic of patients presenting with QFR >0.80 in all the analyzed vessel and those with ≤0.80 in at least one vessel.

	GROUP 1 QFR > 0.80 (131)	GROUP 2 QFR ≤ 0.80 (25)	*p*-Value
Lesion length LAD ± SD, mm (*n*)	18.87 ± 11.25 (97)	22.52 ± 12.34 (25)	0.16
Lesion length RCA ± SD, mm (*n*)	14.67 ± 7.71 (85)	17.39 ± 10.91 (20)	0.20
Lesion length LCX ± SD, mm (*n*)	14.63 ± 8.19 (103)	24.26 ± 15.99 (20)	0.0001
DS% LAD ± SD, (*n*)	33.55 ± 7.64 (97)	46.14 ± 7.31 (25)	<0.0001
DS% RCA ± SD, (*n*)	31.98 ± 7.97 (85)	39.47 ± 11.25 (20)	0.0008
DS% LCX ± SD, (*n*)	33.43 ± 7.64 (103)	41.16 ± 10.00 (20)	0.0001
AS% LAD ± SD, (*n*)	36.10 ± 11.16 (97)	50.43 ± 11.78 (25)	<0.0001
AS% RCA ± SD, (*n*)	36.48 ± 12.12 (85)	46.02 ± 15.47 (20)	0.0034
AS% LCX ± SD, (*n*)	34.99 ± 10.71 (103)	43.76 ± 14.41 (20)	0.002
RVD LAD ± SD, mm (*n*)	2.56 ± 0.65 (97)	2.26 ± 0.58 (25)	0.04
RVD RCA ± SD, mm (*n*)	2.93 ± 0.67 (85)	3.10 ± 0.60 (20)	0.30
RVD LCX ± SD, mm (*n*)	2.53 ± 0.58 (103)	2.37 ± 0.63 (20)	0.26
QFR LAD, (*n*)	0.93 ± 0.05 (97)	0.75 ± 0.10 (25)	<0.0001
QFR RCA, (*n*)	0.97 ± 0.04 (85)	0.92 ± 0.09 (20)	0.0001
QFR LCX, (*n*)	0.97 ± 0.04 (103)	0.91 ± 0.08 (20)	0.0001

RVD: reference vessel diameter; DS%: diameter stenosis%; AS%: area stenosis%.

**Table 3 jcm-09-00220-t003:** Long-term MACCE (major adverse cardiovascular and cerebrovascular events) predictors at univariate analysis.

Parameter	HR	95%CI	*p*-Value
**Age**	**1.02**	**1.00–1.05**	**0.03**
Diabetes	1.42	0.82–2.46	0.21
Family history	0.95	0.56–1.62	0.86
Dyslipidemia	0.92	0.58–1.45	0.72
Arterial Hypertension	1.06	0.60–1.86	0.85
Obesity	1.12	0.69–1.80	0.65
**Sex category**	**2.24**	**1.27–3.94**	**0.005**
Smoke	0.96	0.61–1.51	0.85
Atrial Fibrillation	1.41	0.82–2.42	0.21
Known CAD	2.15	1.30–3.55	0.002
COPD	1.53	0.81–2.90	0.20
Severe LV dysfunction	1.53	0.62–3.79	0.36
**CKD**	**2.27**	**1.09–4.73**	**0.03**
Multivessel CAD	2.55	1.62–4.02	0.0001
**Previous PCI**	**2.03**	**1.28–3.20**	**0.003**
Previous MI	1.51	0.92–2.47	0.11
Severe valvular disease	1.70	0.62–4.66	0.30
ACEi or ARBs	1.22	0.72–2.07	0.47
ASA	1.63	0.95–2.79	0.08
BB	1.58	0.96–2.60	0.07
CCB	1.06	0.63–1.78	0.83
Nitrates	0.93	0.50–1.71	0.81
Statins	1.52	0.94–2.45	0.09
Lesion length LAD	1.01	0.99–1.03	0.36
Lesion length RCA	1.03	1–1.06	0.10
Lesion length LCX	1.02	1–1.04	0.09
QFR LAD ≤ 0.80	3.08	1.73–5.50	0.0001
QFR RCA ≤ 0.80	3.24	1.17–8.97	0.02
QFR LCX ≤ 0.80	-	-	-
RVD LAD	0.72	0.47–1.11	0.14
RVD RCA	1.01	0.65–1.56	0.98
RVD LCX	1.08	0.70–1.67	0.72
AS% LAD	1.03	1.00–1.05	0.02
AS% RCA	1.03	1.01–1.06	0.002
AS% LCX	1.02	1–1.04	0.07
DS% LAD	1.04	1.01–1.07	0.0076
DS% RCA	1.05	1.02–1.08	0.001
DS% LCX	1.03	1–1.06	0.07
QFR LAD	0.01	0.001–0.09	0.0001
QFR RCA	0.001	0.00–0.03	0.0003
QFR LCX	0.003	0.00–0.33	0.02
**≥1-vessel QFR ≤0.80**	**3.42**	**2.06–5.67**	**<0.0001**

COPD: chronic obstructive pulmonary disease; LV: left ventricle; CKD: chronic kidney disease; ASA: Acetylsalicylic Acid; BB: Beta-blockers; CCB: Calcium channel blockers.

**Table 4 jcm-09-00220-t004:** Long-term MACCE predictors at multivariate analysis.

Parameter	HR	95%CI	*p*-Value
Age	1.02	0.99 to 1.04	0.19
Sex	1.44	0.78 to 2.67	0.24
CKD	2.81	1.26 to 6.26	0.01
Previous PCI	1.90	1.15 to 3.15	0.01
Valvular disease	1.14	0.55 to 2.35	0.73
≥1-vessel QFR ≤0.80	3.14	1.78 to 5.54	0.0001

Overall model fit: Chi-Square 42.80, *p* < 0.00001.

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
