# Peer review of "QFR Predicts the Incidence of Long-Term Adverse Events in Patients with Suspected CAD: Feasibility and Reproducibility of the Method"

_jcm, 2020, doi:10.3390/jcm9010220_

Round 1

Reviewer 1 Report

This paper presents valuable results from post-hoc analysis of 167 patients without significant stenoses in aspect of QFR measurement utility to predict MACCE. 

However, I have some concerns that need to be explained:

1) Introduction

As authors mentioned, QFR is currently considered as better method for evaluation of hemodynamic relevance of coronary lesions, so its clinical usefulness in different conditions should be indicated more precisely.

2) Methods

Chronic kidney disease may influence on QFR predictive values. It would be justified to exclude these patients from analysis.

3) Results 

a) One patient has incomplete data, I suggest to exclude he/she from analysis (table 1 will be more clear).

b) How did authors calculate 0.8 QFR value as a "breakthrough" value for patients distribution into 2 groups? 

c) Table 3 - decrease the number of decimal places.

d) What was the predictive power of the model created in multivariate analysis?

4) Discussion

a) I suggest to refer obtained results to values indicated by KoÅ‚towski et al (https://doi.org/10.1007/s00392-018-1258-7) in aspect of assessment of patients with intermediate coronary stenosis and to Biscaglia et al (DOI:10.1016/j.jcin.2019.06.003)  in aspect of prognostic value after stent implantation. 

b) It is very intriguing that nor obesity neither received treatment did not influence on long-term MACCE evidences. Authors should comment this.

General remark:

Group 2 with FR lower than 0.8 has only 25 patients, it can influence on reliability of obtained results.

Authors' Responses to Reviewer's Comments

Introduction

REVIEWER COMMENT: As authors mentioned, QFR is currently considered as better method
for evaluation of hemodynamic relevance of coronary lesions, so its clinical usefulness in
different conditions should be indicated more precisely.

ANSWER: We appreciate this comment. We updated the introduction with the more recent
evidence published in literature concerning the different settings in which QFR might be useful.
The evidence supporting the use of this method is increasing. Its major advantage is that it does
not require hardware (resulting in lower costs) and that it is less invasive (as it does not involve
wiring of the vessel). It is however important to show whether the QFR information also has
prognostic relevance.

Methods

REVIEWER COMMENT: Chronic kidney disease may influence on QFR predictive values. It
would be justified to exclude these patients from analysis.

ANSWER: We agree with this comment and with the potential influent role of CKD concerning
the outcome. We performed an additional sensitivity analysis excluding the patients with kidney
failure (now reported in the “secondary endpoints” paragraph). This analysis confrmed the
validity of the observation from the general population.

Results

REVIEWER COMMENT: One patient has incomplete data, I suggest to exclude he/she from
analysis (table 1 will be more clear).

ANSWER: Thank you for the comment. We analysized the clinical report at the moment of
patient’s enrollment in the Flow-Mec Database and we completed the missing information. We
also updated Table 1. Please have look to the new calculated percentages. We replace the updated
percentages in the text as well.

REVIEWER COMMENT: How did authors calculate 0.8 QFR value as a "breakthrough" value
for patients distribution into 2 groups?

ANSWER: Thanks for the comment. 0.80 is the standard threshold of positivity for FFR,
validated by a number of studies including large randomized controlled trials. You are right that
this information is still missing for QFR (which was however developed using FFR as
benchmark). For this reason, we provided ROC curves identifying the sensitivity and specificity
of other threshold levels, and we also performed an analysis for a QFR<0.90. Nonetheless, for the
sake of data presentation we used QFR<0.80 because this is the most validated threshold. The
population has been divided in two groups according QFR values calcutated in all available
vessel in each patient: patients with QFR>0.80 in all analyzed vessels and patients with
QFR≤0.80 in at least one vessel. In order to make clearer this distinction we modified the title of
Table 1 and Table 2.

REVIEWER COMMENT: Table 3 - decrease the number of decimal places.

ANSWER: Done. Thanks for the suggestion.

REVIEWER COMMENT: What was the predictive power of the model created in multivariate
analysis?

Thank you very much for this question. The overall model fit was evaluated with Chi-Square,
which provides an estimate of the relationship between time and all the variables in the model.
The value was 42.80, P<0.00001. This is now reported in the legend to Table 4.

Discussion

REVIEWER COMMENT: I suggest to refer obtained results to values indicated by Kołtowski et
al (https://doi.org/10.1007/s00392-018-1258-7) in aspect of assessment of patients with
intermediate coronary stenosis and to Biscaglia et al (DOI:10.1016/j.jcin.2019.06.003) in aspect
of prognostic value after stent implantation.

ANSWER: Thanks for the comment. We appreciated the articles you recommended us and we
introduced them in our paper. We have cited the articles (ref number 13 Kołtowski et al and number 17 Biscaglia et.) in the introduction section following your previous suggestion to improve the introduction considering a broader spectrum of QFR indications.

REVIEWER COMMENT: It is very intriguing that nor obesity neither received treatment did not
influence on long-term MACCE evidences. Authors should comment this.

ANSWER: We agree with reviewer’s comment and we discussed the issue in section
“discussion”. However, among limitations we reported that “Finally, adherence to medical
therapy during follow-up was not assessed”, and this could in part explain your shareable
comment.

General remark

REVIEWER COMMENT: Group 2 with FR lower than 0.8 has only 25 patients, it can influence
on reliability of obtained results.

ANSWER: Thanks for the comment. Since the studied population consisted of patients with
coronary artery stenoses judged angiographically non-significant, a lower number of positive offline
QFR is expected (would be suprising the opposite finding). However, patients with positive
QFR (25) represent 16% of the entire population (156), and this could reflect into clinical
practice, suggesting the need of a more routinarly use of functional analyses for intermediate
coronary stenoses. We have considered this comment and we pointed it out in the discussion
section.

Round 2

Reviewer 1 Report

Authors improved manuscript significantly. 

Reviewer 2 Report

The paper is interesting and well written. Methods,results and conclusion are optimally descibed and also limits of the research.Bibliography is up to date.